# Making Large Language Models Better Data Creators

**Dong-Ho Lee[1][*], Jay Pujara[1], Mohit Sewak[2], Ryen W. White[2], Sujay Kumar Jauhar[2]**

[1]Information Sciences Institute, University of Southern California
[2]Microsoft Research

{dongho.lee}@usc.edu, {jpujara}@isi.edu, {mohit.sewak,ryenw,sjauhar}@microsoft.com

## Abstract

Although large language models (LLMs) have advanced the state-of-the-art in NLP significantly, deploying them for downstream applications is still challenging due to cost, responsiveness, control, or concerns around privacy and security. As such, trainable models are still the preferred option in some cases. However, these models still require human-labeled data for optimal performance, which is expensive and time-consuming to obtain. In order to address this issue, several techniques to reduce human effort involve labeling or generating data using LLMs. Although these methods are effective for certain applications, in practice they encounter difficulties in real-world scenarios. Labeling data requires careful data selection, while generating data necessitates task-specific prompt engineering. In this paper, we propose a unified data creation pipeline that requires only a single formatting example, and which is applicable to a broad range of tasks, including traditionally problematic ones with semantically devoid label spaces. In our experiments we demonstrate that instruction-following LLMs are highly cost-effective data creators, and that models trained with these data exhibit performance better than those trained with human-labeled data (by up to 17.5%) on out-of-distribution evaluation, while maintaining comparable performance on in-distribution tasks. These results have important implications for the robustness of NLP systems deployed in the real-world.

## 1 Introduction

Large language models (LLMs) have revolutionized the field of NLP, yielding impressive performance on various conventional natural language understanding (NLU) and generation (NLG) tasks. They are able to do this with only a handful (*i.e.,* few-shot) or sometimes even no training examples (*i.e.,* zero-shot) (Brown et al., 2020; Du et al., 2022;

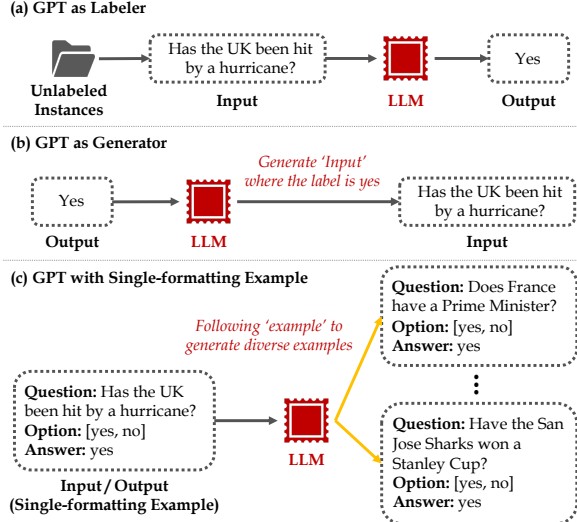

Figure 1: Existing LLM-based data augmentation needs unlabeled examples (labeler) or label-specific prompts (generator), while our framework generates examples for a variety of tasks in a unified way.

Rae et al., 2021; Thoppilan et al., 2022; Chowdhery et al., 2022). However, despite their effectiveness, there is a continued demand for the deployment of smaller trainable or tunable models in real-world scenarios due to cost constraints, existing service-level agreement response times, or privacy and security concerns around using black-box APIs. Unfortunately, application-specific custom models sometimes require large amounts of high-quality human-labeled data, in order to perform well. Thus, in order to reduce time and cost in the model deployment cycle, recent work has focused on trying to obtain training data by leveraging LLMs as either **labelers** to annotate unlabeled data (Yoo et al., 2021; Wang et al., 2021a; Lang et al., 2022), or **generators** to generate new data samples (Meng et al., 2022; Ye et al., 2022; Gao et al., 2022).

Despite initial successes, constraints for these techniques continue to hinder their applicability in broader real-world settings. First, in the context of using LLMs as **labelers**, it is essential to have raw

---
[*]Work done during Microsoft Research Internship.

data that closely resembles the distribution of data in the predictive task. Most previous research has assumed access to a training dataset from which the labels are elided; however, for cold-start problems in the real-world, no such assumptions can be made. Curating raw data for tasks in specialized domains, such as those in the biomedical or legal fields, can be particularly challenging. Conversely, sampling a large volume of data at random can result in an imbalanced label distribution due to rare events (Markov et al., 2022).

Meanwhile, leveraging LLMs as **generators** requires careful curation of few-shot examples (Hartvigsen et al., 2022), or composition of prompts that highlight the semantic meaning of labels (Wang et al., 2021b; Meng et al., 2022; Ye et al., 2022; Gao et al., 2022), such as *positive* v. *negative* in sentiment classification. The latter has been a bottleneck to the broader applicability of LLMs as generators, however, since not all tasks have labels that are semantically meaningful, or are enumerable. Consider, for example the label *yes* v. *no*, which have no meaning when taken without context; or the options of a multiple choice QA (see Figure 1), which are an effectively open-ended label-set that varies from instance to instance. For these kinds of problems LLMs as **generators** continue to be inadequate.

In this paper, we first present a formal framework for characterizing different approaches for LLM data creation. Specifically, we use graphical models as a way to characterize and unify disparate approaches that include LLMs as either **labelers** or **generators** (Section 2). Next, we propose a novel data creation pipeline that only requires a single formatting example to generate heterogeneous labeled data for various downstream applications, including those that focus on specialized domains. In contrast to current methods that require dataset-specific components (*e.g.,* label description, example selection), our pipeline serves as a unified solution that can be applied to a wide range of tasks, including those where the label set is either semantically devoid of meaning, or unenumerable.

Our data creation pipeline leverages an instruction-following LLM as a **generator** in conjunction with a single formatting example as a simple yet effective way of imposing structured constraints. Specifically, our approach iteratively conditions the generator on an instruction and a unique formatting example in a JSON format to

yield multiple examples that vary in content but are formatted uniformly (Section 3.1−3.2). Furthermore, as an efficient means of diversifying the generated data, we propose a "self-reference" strategy, which iteratively samples from the pool of newly created examples to seed the prompt for the next round of generation (Section 3.4). Specifically, we outline 4 distinct instantiations of "self-reference" including **random**, **contrastive**, **similar**, and **tree** sampling for controlled diversification of data.

We evaluate our data creation pipeline on a battery of tests involving three distinct types of tasks, namely multiple-choice question answering (QA), open-book yes/no QA, and closed-book yes/no QA. The datasets for these tasks range across a variety of domains, including specialized ones such as the biomedical domain. Furthermore, for each category of task, we use a minimum of two datasets in order to compare the out-of-distribution (OOD) generalization of models using original data to synthetically generated LLM data. Our results demonstrate that leveraging LLMs as generators using our formatting-based creation approach is a highly cost-effective way of creating data that can be effectively used to train models for a variety of downstream tasks, including those in specialized domains, and ones where labels are devoid of semantic meaning or vary across the data. For in-distribution (ID) settings, naturally having access to large amounts of high-quality manually curated and labeled data is still ideal. However, when only a small amount of human-labeled data is available, our approach yields results that are often comparable, and sometimes even better than the original datasets. This highlights the potential role LLMs can play in the model development cycle, especially in resource-poor or specialized domains. Further, for the OOD settings, models trained on data generated by our pipeline consistently, and by large margins, outperform their counterparts trained on data from human sources. This robustness and generalizability has important implications for the deployment of real-world systems that deal with data that are variable, chaotic and often very different from curated academic datasets. We are realeasing our code and prompts to the community to spur future research in the area[1].

---

[1]https://github.com/microsoft/llm-data-creation

## 2 Formalization of LLM-based data creation

In this section, we attempt to draw different data creation strategies using LLMs into a unified framework, and discuss related research using this framework.

### 2.1 LLM-based data creation

Assume a large language model $\mathcal{M}$ (*e.g.,* GPT-3) that has been pre-trained to maximize the likelihood of generating each token in a sequence $\mathbf{x} = [x_1, x_2, \ldots, x_n]$ by conditioning on previous tokens. Then, $\mathcal{M}$ is capable of generating new text through recursive sampling of tokens from its output probability distribution. Given such a model $\mathcal{M}$ and label space $\mathcal{Y}$, the goal of data creation is to induce samples $(\mathbf{x}, \mathbf{y})$ where $\mathbf{y} \in \mathcal{Y}$. Based on different instantiations of this general framework, other inputs may be included, such as a label descriptive prompt $\mathcal{W}_\mathbf{y}$ for each $\mathbf{y} \in \mathcal{Y}$, in-domain unlabeled example $\mathbf{x}_u \in \mathcal{D}_U$, or a small number of example pairs $(\mathbf{x}_l, \mathbf{y}_l) \in \mathcal{D}_L$ along with their corresponding explanation $\mathbf{e}_l$.

### 2.2 Formal Framework and Related Work

Given these basic components, there are two broad strategies for LLM data creation, namely using an LLM as a labeler or as generator. Graphical models for each of these two strategies is presented in Figure 2 to summarize the conditional interactions and independence assumptions that describe the relationships between common framework constituents. The rest of this section discusses existing work using the unified language of these graphical models.

**Using LLMs as labelers.** $\mathcal{M}$ can be used as a labeler for unlabeled data (See Figure 2 (a)). Here, approaches assume that unlabeled data $\mathcal{D}_U$ is provided as input and the distribution of $\mathcal{D}_U$ is similar to the distribution for the target task. Labeling can be achieved either by conditioning $\mathcal{M}$ on a few labeled examples $(\mathbf{x}_l, \mathbf{y}_l) \in \mathcal{D}_L$ (Brown et al., 2020; Yoo et al., 2021; Wang et al., 2021a; Lang et al., 2022), or by leveraging instructive prompts $W$ describing the task without any labeled examples (Brown et al., 2020). When providing the model $\mathcal{M}$ with a small number of labeled examples $(\mathbf{x}_l, \mathbf{y}_l) \in \mathcal{D}_L$, – often referred to as the *few-shot* setting (and contrasted with the *zero-shot* setting, where no examples are provided) – recent studies

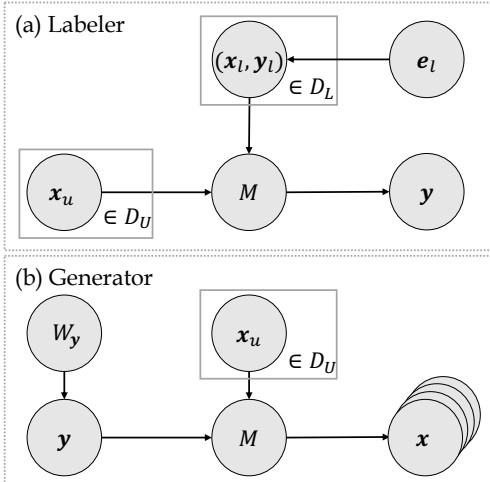

Figure 2: **Graphical models** for using LLM $\mathcal{M}$ as **(a) labeler** which outputs label $\mathbf{y}$ for unlabeled data $\mathbf{x}_u \in \mathcal{D}_U$ using instructive prompt $\mathcal{W}_\mathbf{y}$ or few-shot examples $(\mathbf{x}_l, \mathbf{y}_l) \in \mathcal{D}_L$ with or without explanation $\mathbf{e}_l$, and as **(b) generator** which generates multiple data $\mathbf{x}$ for label $\mathbf{y}$ with label-descriptive prompt $\mathcal{W}$ or using in-domain unlabeled example $\mathbf{x}_u \in \mathcal{D}_U$ .

have shown that curating diverse and representative samples is critical to the ability of the model to label new samples (Liu et al., 2022; Rubin et al., 2022; Su et al., 2022). This can be challenging, particularly in specialized domains such as, for example, the legal or biomedical domain – where only a small number of samples may be curated. Our paper proposes a pipeline (Section 3 capable of tackling these challenges, and is particularly useful in resource-poor domains (Section 5.1). Furthermore, providing intermediate reasoning steps (*i.e.,* chains-of-thought) as explanations $\mathbf{e}_l$ in prompts enables better labeling in both few-shot (Wei et al., 2022; Zhou et al., 2022; Lampinen et al., 2022) and zero-shot setting (Kojima et al., 2022).

**Using LLMs as generators.** An altogether different data creation approach uses the LLM $\mathcal{M}$ directly as generator (See Figure 2 (b)). While a labeler predicts $\mathbf{y}$ by conditioning on an input $\mathbf{x}$, a generator does the reverse by generating $\mathbf{x}$ given $\mathbf{y}$. However, like with LLMs as labelers, a small number of relevant samples can be used to condition $\mathcal{M}$ for generating data. Hartvigsen et al. (2022) feeds human-curated examples of the target label (*e.g.,* implicit hate speech) into $\mathcal{M}$ to generate human-like examples for the target label. In contrast, Wang et al. (2021b) conditions $\mathcal{M}$ on both in-domain unlabeled examples and target label $\mathbf{y}$ to generate domain-related data for $\mathbf{y}$. Meanwhile a number of different efforts (Meng et al., 2022;

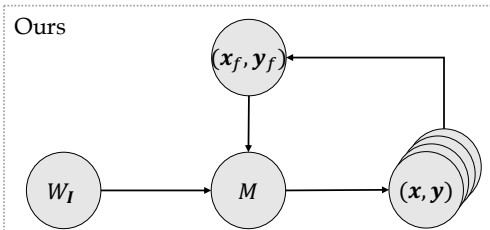

Figure 3: **Framework Overview** of example-based data creation which outputs multiple complete data $(\mathbf{x}, \mathbf{y})$ starting with an initial seed formatting example $(\mathbf{x}_f, \mathbf{y}_f)$ and the instruction $\mathcal{W}_\mathbf{I}$.

Ye et al., 2022; Gao et al., 2022) condition $\mathcal{M}$ on in-domain unlabeled example and well-formatted descriptive prompts $\mathcal{W}_\mathbf{y}$ for target label $\mathbf{y}$ to generate data. One important caveat with all these approaches is that the label $\mathbf{y}$, upon which outputs are conditioned, needs to be inherently meaningful in order for instructions to be formulated in a way that prompt the model $\mathcal{M}$ into generating coherent outputs. For example, when $\mathbf{y}$ is an entailment relationship, a corresponding prompt $\mathcal{W}_\mathbf{y}$ might include "$\mathbf{x}_u \in \mathcal{D}_U$. *In other words...*"; or when $\mathbf{y}$ is the sentiment of a movie review, $\mathcal{W}_\mathbf{y}$ might include "*The movie review is...*". Contrast this with the scenario where $\mathbf{y}$ is an index or binary response, which has no semantic meaning without context and is therefore difficult to condition on. In this paper, we devise a unified approach to tackling these scenarios (Section 3), yielding a method for data creation that is broadly applicable.

**Differences with Instruction-following Data Generation.** Recent studies use $\mathcal{M}$ to create data for training instruction-following models (Wang et al., 2023b; Taori et al., 2023; Xu et al., 2023; Chiang et al., 2023; Mukherjee et al., 2023). Such studies leverage $\mathcal{M}$ as a labeler, producing a coherent response $y$ for a given instruction $x$. The main focus is on covering a wide range of inputs $x$ and their corresponding responses $y$ to encompass the diversity of instructions potentially encountered in real-world interactions with users. While recent studies have indicated that such models trained with auto-generated instruction-response pairs can yield logical and coherent responses to user instructions, their performance on NLU tasks remains sub-par (Wang et al., 2023a). Meanwhile, the focus of our work is the generation of data specifically tailored for natural language understanding (NLU) tasks with a focus on accurate responses.

| Instruction |
| --- |
| - You are creating {*number_of_examples*} examples that follow the format of the example provided, but with a different content. |
| - The created examples **must** all have different answers. |
| - The output **must** be in unnumbered JSON format. |
| - [*fixed_only*] The created examples **must** have the same options as the provided example. |

Table 1: **Instruction** $\mathcal{W}_\mathbf{I}$ used in the paper.

## 3 Example-based Data Creation

This paper proposes a unified data creation approach using LLMs, which does not need in-domain unlabeled examples $\mathbf{x}_u \in \mathcal{D}_U$ or data-specific label-descriptive prompts $\mathcal{W}_\mathbf{y}$. As illustrated in Figure 3, our framework iteratively creates data $\mathcal{D}_G$ beginning with a single initial formatting example $(\mathbf{x}_f, \mathbf{y}_f)$ and an instruction $\mathcal{W}_\mathbf{I}$ (Section 3.1). The process begins by converting $(\mathbf{x}_f, \mathbf{y}_f)$ into a structured prompt $\mathcal{W}_f$ (Section 3.2-3.3). After conditioning $\mathcal{M}$ on $[\mathcal{W}_I; \mathcal{W}_f]$ to generate data, we sample an instance from the pool of newly created data to serve as formatting example for the next iteration (Section 3.4). We continue to iterate in this manner until the required number of instances $k$ is obtained, after discarding duplicate and ill-formatted data. This is done by caching data and checking newly created candidates against previously generated ones for duplicates, and by using the python `json.loads()` method for verifying the validity of the json output. The resulting data creation pipeline is generally suitable for most classification tasks. Although, in this paper we specifically focus on tasks where the label set is potentially open-ended (*e.g.,* multiple-choice QA), or lacks inherent semantic meaning (*e.g.,* binary QA) – problem spaces that have posed challenges to past work in LLM data creation.

### 3.1 Instruction

The goal of our framework is to have the model $\mathcal{M}$ generate a diverse set of examples in the same format as the input formatting example $(\mathbf{x}_f, \mathbf{y}_f)$. To ensure format consistency and example diversity, we use the system instruction $\mathcal{W}_\mathbf{I}$ in Table 1. We generate data in batches of {*number_of_examples*}, not only to account for the token generation limits of LLMs, but also to encourage content diversity through subsequent sampling of $\mathcal{W}_f$ (Sec 3.4). In this paper, we set {*number_of_examples*} to 5 and do not vary it. In order to mitigate label bias, we encourage mod-

els to strive for maximal variance in their generated responses, avoiding repetitions in data where the answer is consistently "yes", for example.

## 3.2 Formatting Example

The only assumed input to our example-based data creation pipeline is a single formatting example $(\mathbf{x}_f, \mathbf{y}_f)$ and its corresponding label space $\mathcal{Y}$. This example is formatted as a JSON-structured prompt $\mathcal{W}_f$ as shown in Figure 4. Given the one-shot JSON structured format prompt, it is expected that the model yields a syntactically correct output that conforms to the JSON schema. While generating a complex structured output like JSON can be challenging, its easy parsing acts as a way to validate outputs at creation time and for effective usage in training downstream models.

## 3.3 Structure of Formatting Example

Recall that our focus in this paper is on data creation for tasks that are challenging because their output label space is open-ended, or because they lack inherent semantic meaning. We refer to these distinct settings using the shorthand *variable* and *fixed*, and note that the input formatting example $(\mathbf{x}_f, \mathbf{y}_f)$ is different for each of these label space settings. Specifically, the major difference is the order of presentation of prompt components.

**Variable Option.** The variable option format is structured in a logical sequence beginning with the question $\mathbf{x}_f$, followed by a list of answer candidates $\mathcal{Y}$, and finally the correct answer $\mathbf{y}_f$.

**Fixed Option.** In contrast, for the variable option, the expected format consists of the answer candidates $\mathcal{Y}$ first, followed by the correct answer $\mathbf{y}_f$, and finally the question $\mathbf{x}_f$. This inversion of prompt components is added to ensure that the auto-regressive model creates questions with pre-determined optionssince the model, as a free generator, can produce inconsistent output, resulting in answer options $\mathcal{Y}$ that do not belong to the fixed pre-determined set.

## 3.4 Self-Reference

Relying on a single formatting example $(\mathbf{x}_f, \mathbf{y}_f)$ as a reference point for all iterations of data creation may limit the ability of the pipeline to yield data that is broad coverage, diverse and balanced. To overcome this, we propose "self-reference", wherein the formatting example $\mathbf{f}_i = (\mathbf{x}_{f_i}, \mathbf{y}_{f_i})$ for all subsequent generation steps $i > 0$ are sampled

```
{
    "Question": "I am black when you buy me, red when you
                use me. When I turn white, you know it's time
                to throw me away. What am I?",
    "Options": ["charcoal", "rose flower", "ink", "fruit", "shoe"],
    "Answer": "charcoal"
}
```

**(a) Variant (multiple-choice QA)**

```
{
    "Options": ["yes", "no", "maybe"],
    "Answer": "yes",
    "Question": "Is batman and robin a sequel to batman
                forever?",
    "Context": "With the box office success of Batman Forever
                in June 1995, Warner Bros. immediately
                commissioned a sequel. …"
}
```

**(b) Fixed (yes-no QA)**

Figure 4: **Example of formatting example prompt** $\mathcal{W}_f$ where "options" contain the label space $\mathcal{Y}$ of the task, "answer" contains $\mathbf{y}_f$ and "question" contains $\mathbf{x}_f$. "Context" is optional and depends on the task.

from the outputs $(\mathbf{x}_{g_{i-1}}, \mathbf{y}_{g_{i-1}}) \in \mathcal{D}_{G_{i-1}}$ generated at iteration $i-1$. We experiment with four different sampling strategies.

**Random selection.** During each iteration, a formatting example for the next step is randomly chosen from the output of the current step.

**Contrastive selection.** For each iteration, we select the example that displays the greatest semantic contrast to the preceding formatting example. In this approach, we use a pre-trained bidirectional-encoder (Reimers and Gurevych, 2019) to generate embeddings for examples, and compute the cosine similarity between $\mathbf{x}_f$ and $\mathbf{x}_{g_{i-1}}$, selecting the instance $\mathbf{x}_{g_{i-1}}$ with the lowest similarity.

**Similar selection.** This sampling approach works analogously to *Contrastive selection*, except that instead of selecting the $\mathbf{x}_{g_{i-1}}$ with the lowest cosine similarity to $\mathbf{x}_f$, we select the one with the highest similarity.

**Tree selection.** Iterative sampling of data may result in significant domain drift from the first seed example to data generated in later steps of the generation pipeline, due to unexpected content variations produced by the model. To avoid this issue, we use all the generated outputs from one step as formatting examples for subsequent iterations. This approach can be viewed as a breadth-first tree traversal over generated examples, and is in contrast with the other three sampling approaches that use a depth-first exploration strategy. Our hypoth-

esis is that the minimum height of the exploration tree yields samples that are more topically coherent.

## 4 Experimental Setup

In this section we describe the experimental setup that we use to evaluate our single-shot example-based data creation framework.

### 4.1 Datasets

We evaluate on three types of different tasks: multiple-choice question answering (QA), open-book yes/no QA, and closed-book yes/no QA – as shown in Table 2. Multiple-choice QA is used evaluate our data creation pipeline in a variable label space setting, while the other tasks are used for fixed label space settings. In order to demonstrate the domain generalization capability of our approach, we additionally use a minimum of two datasets for each category of task. The datasets range broadly in the reasoning abilities they demand from models, requiring them to solve diverse problems such as filling in the blank in a sentence (PIQA (Bisk et al., 2020), WinoGrande (Sakaguchi et al., 2021)), choosing the most suitable option among multiple choices (CommonsenseQA (Talmor et al., 2019), RiddleSense (Lin et al., 2021)), comprehending a given passage to make a prediction (BoolQ with context (Clark et al., 2019), PubMedQA (Jin et al., 2019), BioASQ (Tsatsaronis et al., 2015)), and answering based on inherent knowledge (BoolQ without context (Clark et al., 2019), StrategyQA (Geva et al., 2021), CREAK (Onoe et al., 2021)). Details of the various datasets are presented in Appendix A.1.

### 4.2 Evaluation Details

In order to demonstrate the efficacy of our data creation framework, we present a comparison of a downstream model when it is trained on (1) the original train dataset denoted by $\mathcal{D}_L$; and (2) an LLM created dataset, denoted by $\mathcal{D}_G$, where a single seed formatting example is randomly selected from $\mathcal{D}_L$. We are unable to conduct a comparative analysis with other LLM-based data generation methods as they do not provide solutions or prompts engineered for the tasks listed in Table 2. The base model used in this paper is RoBERTa-large (Liu et al., 2019).

| Task | Label Space | Domain | Dataset |
|---|---|---|---|
| multiple-choice QA | Variant (2) | Commonsense | PIQA
Winogrande |
| multiple-choice QA | Variant (5) | Commonsense | CommonsenseQA
RiddleSense |
| open-book yes/no | Fixed (2) | Knowledge
Biomedical
Biomedical | BoolQ (w/ context)
PubMedQA
BioASQ |
| closed-book yes/no | Fixed (3) | Knowledge
Knowledge
Knowledge | BoolQ (w/o context)
StrategyQA
CREAK |

Table 2: **Datasets used in the paper.** The numbers enclosed in parentheses indicate the number of labels within the label space.

### 4.3 Implementation Details

Throughout the entire process of data creation, we use gpt-3.5-turbo language model, as of June 2023, with specific settings of $temperature$ and $top_p$ set to 1. When conducting fine-tuning experiments, we leverage the Adam optimizer (Kingma and Ba, 2014) with a maximum sequence length of 256. In each experiment, we perform a grid search on development data for the optimal learning rate in [3e-4, 1e-4, 5e-5, 2e-5, 1e-5, 5e-6, 3e-6, 1e-6, 5e-7], and batch size in [4, 8, 16]. All experiments are conducted on an RTX A5000 with FP32.

## 5 Experimental Results

We conduct a comprehensive set of experiments that seek to evaluate the effectiveness of data created by LLMs for focused downstream application modeling. Our first evaluation involves examining the performance of models trained on data generated by variants of our single-shot data creation pipeline and comparing them against manually curated training data. We investigate both in-distributed (ID) and out-of-distribution test data settings, each of which are explained in greater detail below.

### 5.1 Performance Comparison

**ID Performance.** In the ID setting the test data is drawn from the same distribution as the training data; specifically, a portion of the full human-sourced dataset is held out as a test set to evaluate models trained on either human-labeled or LLM created data. Table 3 summarizes the performance of models trained on two types of datasets: the original dataset, denoted by $\mathcal{D}_L$, and datasets created using the different "self-reference" variants of our single-shot pipeline – the corresponding rows in the table marked as $\mathcal{D}_G$. In both settings, and for a

| Trained on ↓ | MCQA (2) | | MCQA (5) | | Open Yes/No | | | Closed Yes/No | | |
|---|---|---|---|---|---|---|---|---|---|---|
| | PIQA | WinoGrande | CommonsenseQA | RiddleSense | BoolQ | PubMedQA | BioASQ | BoolQ | StrategyQA | CREAK |
| # Examples in $\mathcal{D}$ | 14,113 | 160 | 8,500 | 3,510 | 9,427 | 450 | 670 | 9,427 | 2,061 | 10,176 |
| $\mathcal{D}_L$ | 80.95 | 51.41 | 68.17 | 56.48 | 85.62 | 55.20 | 87.14 | 65.68 | 49.56 | 81.19 |
| $\mathcal{D}_G$ (Random) | 66.20 | 51.26 | 42.06 | 37.85 | 68.99 | 59.80 | 80.71 | 52.23 | 53.04 | 67.93 |
| $\mathcal{D}_G$ (Contrastive) | 66.15 | 52.36 | 41.57 | 38.43 | 66.66 | 59.20 | 67.14 | **61.28** | 49.56 | 67.93 |
| $\mathcal{D}_G$ (Similar) | 67.15 | 52.05 | 47.62 | 42.09 | 69.60 | 60.60 | 83.57 | **61.28** | 49.56 | 69.24 |
| $\mathcal{D}_G$ (Tree) | **68.35** | **52.81** | **48.50** | **42.26** | **69.66** | **61.60** | **85.71** | **61.28** | **56.52** | **72.74** |
| $(\mathcal{D}_G - \mathcal{D}_L)/\mathcal{D}_L$ | −18.43% | +2.65% | −40.55% | −33.64% | −22.91% | +10.38% | −1.66% | −7.18% | +12.31% | −11.61% |

Table 3: **ID Performance (Accuracy)** comparison between models trained on original train dataset $\mathcal{D}_L$ (First group) and LLM-created train dataset $\mathcal{D}_G$ (Second group). The optimal variant for data-creation in the second group is shown in **bold**, and the second best is underlined. The third group of the table presents the percentage difference between the best variant in the second group and the first group.

| Train → 
 Trained on ↓ Test → | MCQA (2) | | MCQA (5) | | Open Yes/No | | | | Closed Yes/No | |
|---|---|---|---|---|---|---|---|---|---|---|
| | PIQA 
 WinoGrande | WinoGrande 
 PIQA | CommonsenseQA 
 RiddleSense | RiddleSense 
 CommonsenseQA | BoolQ 
 PubMedQA | PubMedQA 
 BoolQ | BioASQ 
 PubMedQA | PubMedQA 
 BioASQ | StrategyQA 
 CREAK | CREAK 
 StrategyQA |
| $\mathcal{D}_L$ | **52.05** | 44.65 | 41.51 | 40.93 | 62.80 | 58.65 | 67.14 | 56.20 | 49.27 | 48.69 |
| $\mathcal{D}_G$ (Random) | 51.57 | 49.10 | 38.51 | 41.33 | 59.00 | 55.77 | 66.42 | 59.40 | 49.27 | 48.69 |
| $\mathcal{D}_G$ (Contrastive) | 50.31 | 49.50 | 32.94 | 42.35 | 59.00 | 59.87 | 75.00 | 55.20 | 49.27 | 46.95 |
| $\mathcal{D}_G$ (Similar) | 48.42 | **52.25** | **43.42** | 42.62 | **64.60** | **62.50** | 77.85 | **63.00** | 49.27 | 51.30 |
| $\mathcal{D}_G$ (Tree) | 50.31 | 49.55 | 40.09 | **43.35** | **64.60** | 61.28 | **81.42** | 66.00 | **57.72** | **54.78** |
| $(\mathcal{D}_G - \mathcal{D}_L)/\mathcal{D}_L$ | −0.93% | +14.54% | +4.39% | +5.58% | +2.78% | +6.16% | +17.53% | +14.84% | +14.63% | +11.11% |

Table 4: **OOD Performance (Accuracy)** comparison between models trained on original train dataset $\mathcal{D}_L$ (First group) and LLM-created train dataset $\mathcal{D}_G$ (Second group). The best dataset for each OOD experiment is shown in **bold**, and the second best is underlined. The third group of the table presents the percentage difference between the best variant in the second group and the first group.

given task, the number of samples for each dataset is identical. We also compute the percentage differential between the two types of datasets, shown in the table as $(\mathcal{D}_G - \mathcal{D}_L)/\mathcal{D}_L$. These findings confirm that while there is no substitute for large amounts of hand-crafted data, – demonstrated by drops of up to −40.55% when using synthetically created data – LLMs can play an important role when access is only available to very little data, and in specialized domains. This is demonstrated by the similar or often better performance of $\mathcal{D}_G$ models on WinoGrande, PubMedQA, BioASQ and StrategyQA. Meanwhile, a comparison between different "self-reference" sampling strategies demonstrates the importance of mitigating domain drift in our single-shot approach, where the sole *true* formatting example is the only anchor point to the data distribution we seek to generate. The **Tree**-based exploration strategy limits the semantic distance between the seed sample and instances created later in the generation process and therefore yield higher performance on ID data.

**OOD Performance.** While ID data is useful to gain insights of a system in controlled settings, real-world applications must deal with data that is often far more variable and chaotic. Therefore, we compare manually curated data (*i.e.,* original training data) to LLM generated data in an OOD setting.

Specifically, since we have used at least two evaluation datasets for each category of downstream application, we use one dataset for training and evaluate on a different dataset. Note that in this setting, while the training data can either be manually curated (i.e. $\mathcal{D}_L$), or generated by an LLM (i.e. $\mathcal{D}_G$), the test dataset is always manually curated (*i.e.,* original test data). Table 4 presents a comprehensive analysis of the OOD performance of models trained on $\mathcal{D}_L$ and $\mathcal{D}_G$. The results show that models trained on LLM data are consistently and sometimes significantly better at OOD predictive performance than their hand-crafted counterparts. This has important implications for the robustness and generalizability of real-world systems that often deal with inputs that are very different from carefully curated academic datasets. We note that a combination of human and LLM created data may yield even higher gains in OOD performance and leave a comprehensive evaluation of this to future work. Finally, a comparison of "self-reference" strategies on the OOD setting shows that while the **Tree**-based exploration approach is still a consistently strong one, other sampling approaches are sometimes comparable or better. This is understandable since some degree of controlled noise is helpful, and can be viewed as a regularizer, when trying to generalize to OOD test data.

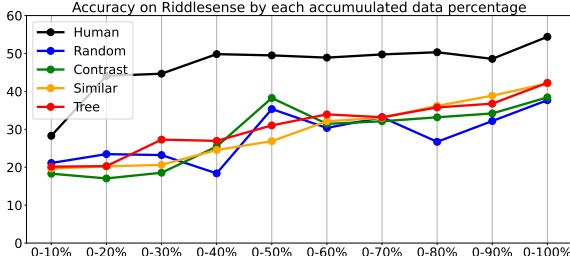

Figure 5: Performance (Accuracy) on RiddleSense using cumulative data splits of the full data.

## 5.2 Distribution shift during creation.

One natural question about the different "self-reference" strategies is whether the domain drift that they induce from iterative sampling detrimentally impacts samples generated later in the creation process. In other words does inclusion of parts of the dataset generated later lead to performance drops or plateaus? In order to answer this question we perform an evaluation of cumulative data splits on one of our benchmark datasets (Riddlesense). Specifically we use incremental percentages of training data – in 10% blocks – for all human labeled and synthetically generated datasets, and evaluate the performance of these models on the corresponding test set. The results of this experiment are shown in Figure 5.

There are several interesting insights to be gained from these results. Firstly, using human-labeled data leads to much faster convergence; this makes sense since the evaluation is performed on ID test data. **Random** and **Contrastive** sampling strategies – both of which performed less well on our main evaluation – do exhibit drops in performance with later cumulative splits. Meanwhile, **Similar** and **Tree** approaches – which were consistently better sampling strategies – demonstrate a steady rise in performance with added data. Jointly these results indicate that judicious selection of examples for subsequent prompts is needed to counter domain drift. Lastly, the final upward trend of all datasets is meaningful because they indicate that models trained on all the datasets do generally benefit from more data. While this additional data is likely difficult and expensive to obtain from human annotators, LLMs can create arbitrarily more data at a fraction of the cost.

## 5.3 Data creation cost

Table 5 presents the expenses incurred by leveraging an instruction-following LLM APIs in our

| Dataset | # Train | Random | Diverse | Similar | Tree |
|---|---|---|---|---|---|
| PIQA | 14,113 | 3.60 | **2.82** | 3.62 | 3.97 |
| WinoGrande | 160 | **0.02** | **0.02** | 0.03 | **0.02** |
| CommonsenseQA | 8,500 | 2.73 | 2.71 | 2.77 | **1.73** |
| RiddleSense | 3,510 | **0.95** | **0.95** | 1.00 | 1.05 |
| BoolQ | 9,427 | 5.13 | **2.24** | 4.95 | 4.2 |
| PUbMedQA | 450 | 0.17 | **0.15** | 0.17 | 0.17 |
| BioASQ | 670 | 0.24 | 0.23 | 0.33 | **0.22** |
| BoolQ | 9,427 | 3.13 | 4.10 | 3.22 | **3.11** |
| StrategyQA | 2,061 | 0.66 | 0.70 | 0.81 | **0.66** |
| CREAK | 10,176 | 3.24 | **3.20** | 4.14 | 3.50 |

Table 5: **API Usage Cost (USD)** of data creation strategy. The cost of utilizing API is calculated in USD, based on the current pricing of `gpt-3.5-turbo` as of June 2023, with a rate of 0.002 USD per 1K tokens. The cheapist strategy is shown in **bold**.

data creation pipeline for each dataset in our evaluation benchmark. The results demonstrate that data creation with LLMs is highly cost-effective, and costs well under $5 USD for every single dataset we created. Factored into this is the cost for data rejected because it was duplicated or ill-formed. Furthermore, our **Tree**-based "self-reference" strategy – which was the most performant on quantitative analyses – was also among the more economical ones. It was the most economical on half the datasets, while the **Contrastive** strategy incurred the lowest cost on the others. These expenses are based on the pricing of `gpt-3.5-turbo` from OpenAI as of June 2023.

## 6 Conclusion

In this paper, we have presented a formal framework for data creation using LLMs and proposed a single-shot formatting example-based data creation pipeline that leverages an instruction-following LLM. Specifically, we showed how multiple varied examples can be generated from a single seed example in a machine-friendly JSON format by conditioning the LLM on a structured prompt consisting of instructions and a formatted example. We further expand the diversity of the output by introducing a "self-reference" mechanism that selects formatting examples for subsequent iterations of generation from newly created data, and present four different instantiations of the sampling strategy. While prior efforts at LLM data creation in the literature have avoided domains where the label space is open-ended and varies from instance to instance, or is semantically devoid of inherent meaning, our structured prompts are able to tackle both. On a battery of evaluations our findings indicate that LLMs can act as highly cost-effective data

generators for the development of small trainable or tunable models in downstream applications. For example, a budget of $5 USD is enough to generate 2M tokens with `gpt-3.5-turbo`, and depending on the task can yield several thousand data samples. These models exhibit noteworthy predictive abilities in generalizing to out-of-distribution data, a key desiderata in real-world systems where data can be messy, variable and evolving. The impact of these findings are meaningful in a number of enterprise scenarios, including for applications that require strict governance on predictive models due to privacy and security concerns, or due to response-time service-level agreements, or indeed for small businesses where human annotation costs, especially from domain experts, can be prohibitively high.

## 7 Limitation

Despite the capability of our pipeline to integrate with a variety of other instruction-following LLMs, our testing is restricted to ChatGPT (*i.e.,* `gpt-3.5-turbo`) due to performance bottlenecks and time-out issues with LLM APIs in general. While we also experimented with several recent open-source instruction-following models that distill ChatGPT responses, their abilities to generate well-formatted JSON and comprehend instructions were limited. We expect that the integration of our pipeline with other open-source LLMs will be possible in time, as the open-source community attains a performance standard commensurate with commercial products.

An important consideration in our single-shot example-based data creation pipeline is the selection of the initial seed formatting sample. We did not perform an exhaustive analysis to understand the impact of this seed selection on data creation quality, again due to performance bottlenecks with LLM APIs. While we select this seed at random in this paper, it is possible that a more carefully considered approach for crafting or selection of this example may yield better results.

## 8 Acknowledgement

This work was funded in part by the Defense Advanced Research Projects Agency (DARPA) and Army Research Office (ARO) under Contract No. N660011924033, Contract No. W911NF-21-C-0002 and Contract No. HR00112390061, and with support from the Keston Exploratory Research Award. The views and conclusions contained herein are those of the authors and should not be interpreted as necessarily representing the official policies, either expressed or implied, of DARPA, ARO or the U.S. Government.

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

# A Appendix

## A.1 Details of Dataset

- **PIQA** (Bisk et al., 2020) is a binary-choice question answering task, which chooses the most suitable solution for questions related to physical commonsense.

- **WinoGrande** (Sakaguchi et al., 2021) is a task that involves selecting the correct binary option to fill in a given sentence that requires commonsense reasoning.

- **CommonsenseQA** (Talmor et al., 2019) is a multiple-choice question answering task, which picks the most appropriate answer on general commonsense questions.

- **Riddlesense** (Lin et al., 2021) is a multiple-choice questions answering task, which picks the most appropriate answer on riddle-style questions that need cognitive process.

- **BoolQ** (Clark et al., 2019) is a question answering task that answering questions with a simple "yes" or "no" response. Questions are naturally occurring queries sourced from the Google search engine. In an open-book setting, the model must comprehend the given context in order to provide an answer, whereas in a closed-book setting, the answer must be provided directly without any context.

- **PubmedQA** (Jin et al., 2019) is a task that involves answering research questions pertaining to the corresponding abstracts of biomedical research papers, and the answers are provided in the form of "yes", "no", or "maybe". In our study, we treat "maybe" as "no" to ensure consistent output format with other datasets.

- **BioASQ** (Tsatsaronis et al., 2015) offers a range of question answering tasks, covering various categories such as factoid, list, summary, and yes/no questions based on the content of biomedical research papers that have been reviewed by experts in the field. For the purpose of this study, our focus will be restricted to questions that have binary answers of "yes" or "no".

- **StrategyQA** (Geva et al., 2021) is a benchmark for question-answering that specifically targets open-domain questions where the necessary reasoning path is not explicitly stated in the question, and needs to be inferred through a strategic approach. The answers to these questions are either "yes" or "no".

- **CREAK** (Onoe et al., 2021) has been specifically formulated for the purpose of commonsense reasoning pertaining to entity knowledge. The dataset comprises assertions of entities, for which the answers need to be specified as either True or False.

## A.2 Data Statistics

| Dataset | # Train | # Valid | # Test |
|---|---|---|---|
| PIQA | 14,113 | 1,838 | 2,000 |
| WinoGrande (XS) | 160 | 633 | 634 |
| CommonsenseQA | 8,500 | 1,221 | 1,241 |
| Riddlesense | 3,510 | 1,021 | 1,202 |
| BoolQ | 9,27 | 1,35 | 1,365 |
| PubmedQA | 450 | 50 | 500 |
| BioASQ | 670 | 75 | 140 |
| StrategyQA | 2,061 | 114 | 115 |
| CREAK | 10,176 | 685 | 686 |

Table 6: **Data statistics.** Each dataset. We use in-house test set which is randomly splitted from the train set for those dataset that do not provide test set.