# OpenReview forum: "Making Large Language Models Better Data Creators"
_EMNLP/2023/Conference — EMNLP 2023 Main_

### Official Review · Reviewer_zQ9a · 2023-07-25

**Typos Grammar Style And Presentation Improvements:** 1. Wrong symbol when citing “Self-gui…
**Soundness:** 4

**Excitement:**

4: Strong: This paper deepens the understanding of some phenomenon or lowers the barriers to an existing research direction.

**Paper Topic And Main Contributions:**

This paper aims to make large language models (LLMs) as better data creators. The authors first unify the frameworks of current LLM data creation methods using the graphical model paradigm. Then, they propose a single yet general format for various, challenging data creations. In addition, self-reference method is added to guarantee the diversity and balance of generation.

**Questions For The Authors:**

A. How to do human evaluation in Figure 5?
B. How to implement "discarding duplicate and ill-formatted data" in section 3?
C. How to avoid the LLM-related faithfulness issue during data creation?
D. What is the optimal setting of your grid search? And are they generally the same in different evaluation dataset?

**Reasons To Accept:**

1. Addressed the important problem about how to use LLM as data creator efficiently. Given the facts that the examples in few-shot are not always available, and the prompting methods in data creation pipeline may require specific label description, they proposed a unified iterative method that is general and efficient, avoiding these two issues
2. Creatively proposed the self-reference method evaluated by random, similarity, or tree search, is added to guarantee the diversity and balance of generation.
3. Solid experiment with various domains under in-distribution (ID) and out-of-distribution (OOD) setting, showing that under the ID setting, such method may be worse in colossal common domain but useful in specialized domain, and the LLM-generated data leads to more robust performance than using the original data in the OOD setting.

**Reasons To Reject:**

1. The paper didn’t provide the comparison to other LLM-based data generation methods. The authors did argue in section 4.2 that it is due to the lack of off-the-shelf prompt template / solution of the tasks listed in this paper. Also, the workload of such comparison seems to be stressful. However, a selling point of this paper is the advantage of their LLM-based data creation method compared to other counterparts, therefore at least some quick, sub-optimal performance comparisons (e.g. by new customized prompt) are desirable.
2. Miss some technique details, e.g. the optimal setting of the grid search (which matters the generalizability of the method), the random seed of the first data selection (which matters the reproducibility), the consistency of such setting (which matters the actual cost since otherwise you need to do grid search each time).

**Reproducibility:**

4: Could mostly reproduce the results, but there may be some variation because of sample variance or minor variations in their interpretation of the protocol or method.

**Reviewer Confidence:**

3: Pretty sure, but there's a chance I missed something. Although I have a good feel for this area in general, I did not carefully check the paper's details, e.g., the math, experimental design, or novelty.

---

> ### Author Rebuttal · Authors · 2023-08-29
>
> We appreciate your feedback on our work. Thank you for taking the time to provide us with valuable insights. Here are our comments on your feedback.
>
> #### **R1. Comparison to other LLM-based data generation**
>
> Thank you for your suggestion, we definitely agree with your comment. Unfortunately, the lack of publicly available prompt templates from related work have made it difficult to yield a fair comparison. We have, in fact, tried to replicate those papers by creating prompts based on their descriptions, but the resulting generated data have generally been poor and difficult to control.
>
> The poor quality could be because: a) we have imperfectly translated the authors’ descriptions to effective prompts; and/or b) because our method is in fact superior. While we would have certainly loved to claim the latter, there was no way for us to tell that this was the only contributing factor, and therefore we decided to not include those results.
>
> However, if it helps to provide context, we are certainly happy to follow your recommendation and include some of the results you have proposed in our final paper.
>
> #### **R2&Q4. Optimal setting of grid search**
>
> We appreciate your question and recognize the importance of clearly explaining our optimal hyperparameter settings. First, we set the `number_of_examples` in a batch (line 289). For this, we found setting to five is the maximum number for efficient generation without encountering formatting errors. Additionally, we utilize temperature and top_p as hyperparameters for the language model (LLM) generation settings. For this, we found that the setting that yields the most randomness (i.e. temperature and top_p set to 1) is the most efficient way to avoid duplication and enables faster generation of the desired number of examples. Finally, the random seed used in our paper for the initial data selection is set to 42.
>
> We will detail and explain all these choices and findings in our final paper.
>
> #### **Q1. Human evaluation in Figure 5**
>
> Model training with human evaluation is training the model using the original training data. In this method, humans are considered as oracle (100% accuracy) and annotate the training data in a sequential manner. To simulate this process, the training data is shuffled, and a certain percentage of the data (N%) is randomly selected for the model training. We will clarify this in the paper.
>
> #### **Q2. Discarding duplicate and ill-formatted data**
> Thanks for your question. We keep track of all previously created inputs and check whether the newly created input has already been previously generated. Also, we filter out ill-formed JSON data by checking whether it can be parsed using the python `json.loads()` function. We will provide more details about these in the final version.
>
> #### **Q3. Faithfulness issue of LLM**
> Thank you for your great question; we agree that LLM faithfulness is a very important issue. For our paper we think it makes sense to break down that measure into two axes: consistency and correctness.
>
> To validate the created data for consistency, we ran an experiment where we asked the LLM to generate predictions on our generated data, and found that those predictions match our generated labels 100%. This indicates that the LLM, whether correct or not, at least maintains label consistency. Meanwhile, correctness of generated outputs is a more challenging metric to gauge, especially at the level of individual data points. However, we contend that downstream performance of the smaller model trained on generated data is a proxy measure of correctness, and as a result our approach - which yields improved results - does generate data that is generally more correct than comparative baselines.
>
> This is, of course, not to say that faithfulness is a solved problem, and there are several ways we could try to improve the quality of our generations. For example, we could add a verification step that checks the factuality and correctness of every generated output as it is produced by the model (at the cost of more LLM API calls), perhaps even augmenting the verification with retrieval from the web to provide supporting evidence. We leave these explorations to future work.
>
> We will make sure to include a detailed discussion of faithfulness in our paper.

---

### Official Review · Reviewer_6qx1 · 2023-08-02

**Typos Grammar Style And Presentation Improvements:** N/A
**Soundness:** 2

**Excitement:**

1: Poor: I cannot identify the contributions of this paper, or I believe the claims are not sufficiently backed up by evidence. I would fight to have it rejected.

**Missing References:**

Wang et al. Self-Instruct: Aligning Language Models with Self-Generated Instructions


**Paper Topic And Main Contributions:**

The paper aims to propose a **data creation** method based on LLM, in particular, `gpt-3.5-turbo`.

The proposed methods can be regarded as an incremental improvement over Self-Instruct (Wang et al.). In particular, the authors improve the "self-reference" by considering contrastive selection, similar selection, and tree selection.

**Questions For The Authors:**

N/A

**Reasons To Accept:**

The authors made some incremental improvements over self-instruct by considering more complicated self-reference strategies including contrastive selection, similar selection, and tree selection.

The distribution shift during data creation is an important issue and is analyzed in this work.

**Reasons To Reject:**

The primary reason to reject is that, the authors proposed an improved variant of self-instruct (Wang et al.) but did not even cite it. It is impossible to consider this as concurrent work with self-instruct so it is unacceptable to miss such an important prior work. It would be okay for me to claim something like "We improve the prior methods and demonstrate its effectiveness on other datasets".

In contrast to the self-instruct paper, which generates diverse tasks, the tasks considered in this work seem rather limited.

**Reproducibility:**

3: Could reproduce the results with some difficulty. The settings of parameters are underspecified or subjectively determined; the training/evaluation data are not widely available.

**Reviewer Confidence:**

4: Quite sure. I tried to check the important points carefully. It's unlikely, though conceivable, that I missed something that should affect my ratings.

---

> ### Author Rebuttal · Authors · 2023-08-28
>
> We appreciate your feedback on our work. Thank you for taking the time to provide us with valuable insights. Here are our comments on your feedback.
>
> #### **R1. Self-Instruct (Wang et al., ACL 2023)**
> While we appreciate the connection you make with the work of Wang et al., and will follow your recommendation to cite them in our paper, we would like to emphasize that our work is different from the task of creating data for training instruction-following LMs (e.g., Self-Instruct). There have been several recent studies that utilize LLMs to generate instruction-answer pairs (e.g., Alpaca, Vicuna, etc.); we highlight the unique aspects of our work and contrast them to that body of work along 3 axes:
>
> - **Purpose**: The main focus of creating instruction-answer pairs (e.g., Self-Instruct) is to generate a diverse set of instructions that people might use in real-world interactions along with coherent responses. The evaluation metric for models trained on this data emphasizes assessing the coherence, helpfulness, and harmlessness of the responses. **However, ensuring correctness is a separate challenge altogether.** Several studies using such instruction-answer pairs have demonstrated poor performance on NLU tasks (See Table 4 in [1] which shows a model trained on Self-Instruct shows the worst performance on MMLU). On the other hand, our primary focus is generating data specifically for NLU tasks. The data we create for NLU tasks must encompass diverse input distributions that are **specific to each task**. At the same time, it is crucial to ensure the **correctness** of the corresponding labels.
>
> - **Methodology**: We contend that our methodology is not an advanced version of the Self-Instruct methodology, but rather a different approach that may share some similarities. (1) Unlike Self-Instruct, which generates instructions and then uses them to generate responses in a two-step process, our approach directly generates responses in a single step; (2) Additionally, Self-Instruct randomly selects 8-shot instruction examples from a pool of 175 instructions, similar to the approach used in ToxiGen [2] mentioned in our paper (line 231) while our work starts with only one seed example and generates data from there.
>
> - **Model**: The main objective of generating instruction-answer pairs (e.g., Self-Instruct) is to train models that have a high number of parameters, like decoder-only or encoder-decoder models, to understand and follow instructions accurately. By using a large parameter size, these models can provide coherent and meaningful responses that reflect their vast knowledge base. However, our goal is to generate training data suitable for smaller models to solve very specific natural language understanding (NLU) tasks effectively. A smaller model does not have the representational capacity of an LLM, and therefore the desiderata for data generation in our scenario are both qualitatively and quantitatively different from a Self-Instruct scenario.
>
> [1] How Far Can Camels Go? Exploring the State of Instruction Tuning on Open Resources., Wang et al., 2023 \
> [2] TOXIGEN: A Large-Scale Machine-Generated Dataset for Adversarial and Implicit Hate Speech Detection., Hartvigsen et al., 2022

---

### Official Review · Reviewer_47zu · 2023-08-05

**Soundness:** 4

**Excitement:**

4: Strong: This paper deepens the understanding of some phenomenon or lowers the barriers to an existing research direction.

**Paper Topic And Main Contributions:**

This work addresses the challenge of utilizing Large Language Models (LLMs) as data generators by presenting a formal framework. Additionally, the paper highlights the advantages of formatting generation tasks in JSON format and introduces several sampling strategies to enhance the diversity of generated examples. The experiments conducted thoroughly analyze the benefits of LLM-generated examples in out-of-distribution (OOD) settings.

**Questions For The Authors:**

1. It would be beneficial to provide the exact prompts used for generating data for the mentioned 10 datasets to facilitate reproducibility and understanding for researchers.
2. Including some examples of LLM-generated data would allow for an evaluation of the quality and usefulness of the generated data.
3. It would be valuable to explore how generation tasks are seamlessly integrated into the proposed framework. Additionally, providing experiment results related to the generation tasks would further support the efficacy of the approach.

**Reasons To Accept:**

1. The paper introduces a formal framework for LLM-based data generation, providing a structured approach.
2. Several mechanisms are designed to improve the quality of generated examples, demonstrating the authors' efforts in enhancing the data generation process.
3. The experiments include a comprehensive analysis of OOD settings and a cost analysis, which contribute to a comprehensive evaluation of the proposed approach.

**Reasons To Reject:**

1. The paper lacks results on training LLMs with the generated examples, which would have provided insights into the effectiveness of the generated data in training LLMs.
2. The focus of the current LLM is primarily on decoder-only LLM's performance on text generation tasks, without specific results on LLM data creation for generation tasks.

**Reproducibility:**

3: Could reproduce the results with some difficulty. The settings of parameters are underspecified or subjectively determined; the training/evaluation data are not widely available.

**Reviewer Confidence:**

4: Quite sure. I tried to check the important points carefully. It's unlikely, though conceivable, that I missed something that should affect my ratings.

---

> ### Author Rebuttal · Authors · 2023-08-28
>
> We appreciate your feedback on our work. Thank you for taking the time to provide us with valuable insights. Here are our comments on your feedback.
>
> #### **R1. Training LLMs with the generated examples**
> While LLMs are extremely capable, they also have some shortcomings (expensive, slow, black-box nature) and are thus not ideally suited to every scenario. Thus the main goal of our paper was to explore methods to leverage the strengths of LLMs for benefitting smaller models that have complementary strengths to LLMs, by generating diverse training data samples to solve natural language understanding (NLU) tasks effectively. The question of using our framework’s generated data to fine-tune an LLM is an interesting proposal that we leave to future work.
>
> #### **R2 & Q3. Data creation for natural language generation (NLG) tasks**
> We definitely agree that natural language generation (NLG) tasks like machine translation and text summarization are crucial, and incorporating LLM-based data creation can be beneficial for these tasks as well. However, the focus of our paper is on natural language understanding (NLU), - specifically where the label-space is semantically devoid - which we believe is, if not more difficult, at least an orthogonal problem for data generation. This is because there is a hard requirement on preserving a strong association between generated text and a correct predictive label, whereas NLG allows for a degree of freedom in generated text. Additionally, there is already a body of work exploring LLM data augmentation for NLG [1-3], whereas its application to NLU with labels that are open-ended or lack semantic meaning (the focus of our work) have been underexplored.
>
> [1] Impossible Distillation: from Low-Quality Model to High-Quality Dataset & Model for Summarization and Paraphrasing., Jung et al., 2023 \
> [2] PLASMA: Making Small Language Models Better Procedural Knowledge Models for (Counterfactual) Planning., Brahman et al., 2023 \
> [3] AUGESC: Dialogue Augmentation with Large Language Models for Emotional Support Conversation., Zheng et al., 2023
>
> #### **Q1&2. Providing exact prompt and examples of generated data**
> We appreciate your suggestion and would like to assure you that the final version will include the exact prompt and examples of the generated data.

---

### Meta-Review · Area_Chair_QLm4 · 2023-09-18

**Recommendation:** 5

**Metareview:**

The paper addresses the challenge of efficiently utilizing Large Language Models (LLMs) as data creators for natural language understanding tasks. It introduces a unified data creation pipeline that requires only a single formatting example and applies to a wide range of tasks. The paper emphasizes the cost-effectiveness of instruction-following LLMs in generating data and demonstrates that models trained with LLM-generated data outperform those trained with human-labeled data on out-of-distribution evaluation while maintaining comparable performance on in-distribution tasks.

Reviewer 1 raised concerns about the lack of results on training LLMs with the generated examples and the paper's focus on NLU tasks. The authors responded by explaining their focus on NLU tasks and leaving the use of generated data for LLM fine-tuning as future work.
Reviewer 3 expressed the need for comparisons with other LLM-based data generation methods, additional technical details, and clarification on various aspects of the paper. The authors addressed these concerns by agreeing to include sub-optimal performance comparisons, provide detailed hyperparameter settings, and explain various processes in the final paper.

**Note:** The review and scores of reviewer "@6qx1" need to be ignored due to the low-quality review.

---

### Decision · Program_Chairs · 2023-10-07

**Decision:**

Accept-Main

**Comment:**

The paper addresses the challenge of efficiently utilizing Large Language Models (LLMs) as data creators for natural language understanding tasks. It introduces a unified data creation pipeline that requires only a single formatting example and applies to a wide range of tasks. The paper emphasizes the cost-effectiveness of instruction-following LLMs in generating data and demonstrates that models trained with LLM-generated data outperform those trained with human-labeled data on out-of-distribution evaluation while maintaining comparable performance on in-distribution tasks.

Reviewer 1 raised concerns about the lack of results on training LLMs with the generated examples and the paper's focus on NLU tasks. The authors responded by explaining their focus on NLU tasks and leaving the use of generated data for LLM fine-tuning as future work.
Reviewer 3 expressed the need for comparisons with other LLM-based data generation methods, additional technical details, and clarification on various aspects of the paper. The authors addressed these concerns by agreeing to include sub-optimal performance comparisons, provide detailed hyperparameter settings, and explain various processes in the final paper.

**Note:** The review and scores of reviewer "@6qx1" need to be ignored due to the low-quality review.